# Efficacy and safety of moderate-intensity rosuvastatin plus ezetimibe versus high-intensity rosuvastatin monotherapy in the treatment of composite cardiovascular events with hypercholesterolemia: A meta-analysis

**Lingyan Liu**[ID][1], **Yongkun Deng**[1☯]*, **Lei Li**[1], **Xingbiao Yang**[1], **Zhaoheng Yin**[1], **Yong Lai**[2,3,4☯]*

1 Department of Medical Protection Center, The 926th Hospital of Joint Logistics Support Force of Chinese People's Liberation Army, Kaiyuan, Yunan, China, 2 Yunnan Provincial Key Laboratory of Entomological Biopharmaceutical R&D, Dali University, Dali, Yunnan Province, China, 3 National-Local Joint Engineering Research Center of Entomoceutics, Dali University, Dali, Yunnan Province, China, 4 College of Pharmacy Dali University, Dali, Yunnan, China

☯ These authors contributed equally to this work.
* 745763076@qq.com (YKD); laiyong8879@163.com (YL)

**Editor:** Jaspinder Kaur, Barking Havering and Redbridge Hospitals NHS Trust: Barking Havering and Redbridge University Hospitals NHS Trust, UNITED KINGDOM OF GREAT BRITAIN AND NORTHERN IRELAND

## Abstract

### Background

Statins are the gold standard in the treatment of dyslipidemia, significantly reducing the risk of cardiovascular disease.

### Objective

To systematically review the efficacy and safety of Moderate-intensity Rosuvastatin Plus Ezetimibe compared with High-intensity Rosuvastatin in treating Composite Cardiovascular Events.

### Methods

PubMed, Embase, Cochrane Library, CINAHL, Web of Science, China Knowledge Network, China Biological Literature Database, Wan Fang Database, and Weipu Database were searched to retrieve randomized controlled trials assessing the safety and efficacy of the two therapies from the time of construction to December 2023. The Jadad scale assessment tool was used to evaluate the quality of the included literature, and Review Manager 5.4 software was used for meta-analysis. The heterogeneity of outcomes was estimated by the $I^2$ test, where we applied risk ratios (RR) and 95% confidence intervals (CI) to assess dichotomous outcomes and mean difference (MD) and 95% CI to present continuous outcomes. We used funnel plots to assess study publication bias and sensitivity analysis was used to address significant clinical heterogeneity.

**Data Availability Statement:** All relevant data are within the paper and its files.

**Funding:** The author(s) received no specific funding for this work.

**Competing interests:** The authors have declared that no competing interests exist.

## Results

The meta-analysis described 21 RCTs involving 24592 participants. The findings indicated that moderate-intensity statin combination therapy improved low-density lipoprotein cholesterol (LDL-C) (MD -8.06, 95% CI [-9.48, -6.64] $p < 0.05$), total cholesterol (TG) (MD -5.66, 95% CI [-8.51, -2.82] $p < 0.05$), and non-high-density lipoprotein cholesterol (non-HDL-C) (MD -17.04, 95% CI [-29.55, -4.54] $p < 0.05$) to a greater extent and superior in achieving LDL-C <70 (RR1.26, 95% CI [1.22, 1.29] $p < 0.05$) and LDL-C <55 (RR1.66, 95% CI [1.56, 1.77] $p < 0.05$) ratios and in the incidence of adverse events than the high-intensity Rosuvastatin monotherapy group. However, there was no statistical difference between the two in improving HDL-C, total cholesterol (TC), and preventing long-term composite adverse cardiovascular events (ACE). Funnel plots indicated publication bias. Sensitivity analysis suggested instability in long-term composite cardiovascular events, HDL-C, and TC results.

## Conclusions

Moderate-intensity statin plus ezetimibe with combination therapy had better efficacy and safety than high-intensity statins. Future validation is needed with more long-term high-quality large samples.

## Introduction

Cardiovascular disease (CVD) is the primary global cause of death and disability and various preventive measures are being pursued to reduce the risk to patients. A primary preventive approach is to control cholesterol, especially LDL-C, to reduce the burden of atherosclerotic plaque, thereby decreasing the likelihood of future cardiac complications. Statins are easy to administer, have few drug interactions, have a favorable safety profile, and are considered the cornerstone in the control of dyslipidemia and atherosclerotic cardiovascular disease (ASCVD) [1–3]. Several statins are available of which rosuvastatin is one of the most effective drugs for reducing CV risk. Unfortunately, muscle symptoms associated with statins have been reported commonly and are a major reason for discontinuing treatment. Moreover, doubling the statin dose resulted in only a 6% increase in LDL-C reduction, but statin-related rhabdomyolysis, hepatic and renal damage, and other adverse effects are also aggravated [4].

To reduce the dosage of statin and improve safety during treatment, more studies, both domestic and international have reported the combination of statin and ezetimibe for lipid adjustment in ASCVD patients, and international guidelines have recommended ezetimibe as a second-line choice for patients who are intolerant to statins or unable to achieve the target LDL-C level [5]. Ezetimibe is a Niemann-Pick C1-Like 1 inhibitor that results in about 15–20% reduction in LDL-C and an increase in HDL-C of about 3%, with no effect on TG. In addition, combination therapy with statins resulted in an extra 21%-27% reduction in LDL-C levels [6,7].

Recently, various randomized controlled trials have shown differences in safety and efficacy between moderate-intensity rosuvastatin combined with ezetimibe and high-intensity rosuvastatin, but there are no consistent conclusions. Kim et al. [8] concluded that Rosuvastatin combination with ezetimibe was more effective, but Choi et al. [9] reported that the advantage was not noticeable. Whether moderate-intensity rosuvastatin combined with ezetimibe has advantages in ACE prevention and lipid regulation needs further exploration. In this study, we

selected the RCTS of medium-intensity Rosuvastatin plus ezetimibe versus double-dose Rosuvastatin for the treatment of CVD patients and performed a meta-analysis on the efficacy and safety of the two groups in order to provide references for the clinical use of drugs.

## Materials and methods

### Data sources and search strategy

We conducted a systematic search of PubMed, Embase, Cochrane Library, CINAHL, Web of Science, China Knowledge Network, China Biological Literature Database, Wan fang Database, and Weipu Database from inception to December 2023. The main search keywords were combinations of "Rosuvastatin", "Ezetimibe", "Moderate-intensity statin", "high-intensity statin" and "randomized controlled trials" in various databases. Furthermore, we manually searched bibliography sections from previously published relevant research to ascertain potential studies.

### Study selection

The search results underwent a title, abstract, and full-text sieve by two reviewers conducted independently, and a third author resolved the differences. Inclusion criteria for this meta-analysis included:(1) age:18-80years;(2) a randomized clinical trial; (3) rosuvastatin 10mg plus ezetimibe vs rosuvastatin 20mg;(4) Included studies had to report at least one clinical event among outcomes of interest;(5) no language restrictions.

### Data extraction and quality evaluation

Two reviewers independently performed data extraction. The extracted information included the first author, year of publication, country, disease status, sample size, interventions, treatment duration, gender, age, and clinical outcomes. A Jadad scale was used to evaluate the quality of the study, which included the generation of randomized sequences (2 scores), allocation concealment (2 scores), blinding (2 scores), and withdrawal and loss of visits (1 score), with 0–2 scores for each of these items according to the criteria. While the total was 7 scores, 1–3 were regarded as low-quality studies, and 4–7 were regarded as high-quality studies.

### Endpoints

Regarding long-term composite ACE, we investigated the primary endpoint (composite of cardiovascular death, major cardiovascular events, or nonfatal stroke), the Secondary efficacy endpoint (composite of all-cause death, major cardiovascular events, or nonfatal stroke), and individual clinical endpoints (all cardiovascular events) in three areas.

Regarding clinical efficacy and safety endpoints, the lipid profile included changes from baseline in LDL-C, HDL-C, non-HDL-C, TG, and TC, as well as the proportion of patients with an LDL-C level of less than 70 or 55 mg/dL; Safety endpoints included the occurrence of overall adverse events (muscle-related adverse events, discontinuation or dose reduction of study drug due to intolerance events or new-onset disease).

### Statistical analysis

All data analyses were performed using RevMan 5.4. We used the risk ratio (RR) and 95% confidence intervals (CI) to evaluate dichotomous outcomes and the mean difference (MD) and 95% CI were used to present continuous outcomes. When original studies do not report the standard deviation of differences for continuous variables, use the Cochrane Handbook to convert the values. The heterogeneity of the results was evaluated by $I^2$ values using the $\chi2$ test.

When heterogeneity showed no substantial differences (P> 0.10, $I^2 < 50\%$) with a fixed effect model, otherwise, the random effects model was employed. There were significant differences between the two groups ($p < 0.05$). We used sensitivity analysis to address the significant clinical heterogeneity. Funnel plots were utilized to assess the research for publication bias when it included more than ten studies.

## Results

### Search results

The target databases, initially turned up 986 pieces of literature for the study, among which 405 were excluded due to duplication, following a review of the titles and abstracts, 240 were eliminated and 280 full-text papers were excluded based on the inclusion and exclusion criteria. Ultimately, the remaining 21 studies (25 data sets) were included in the meta-analysis,13 articles in English and 8 articles in Chinese (Fig 1).

### Study characteristics and quality assessment

The meta-analysis included 21 RCTs with 24592 participants conducted from 2016 to 2023. Patients with various cardiovascular disease statuses were included, such as those with

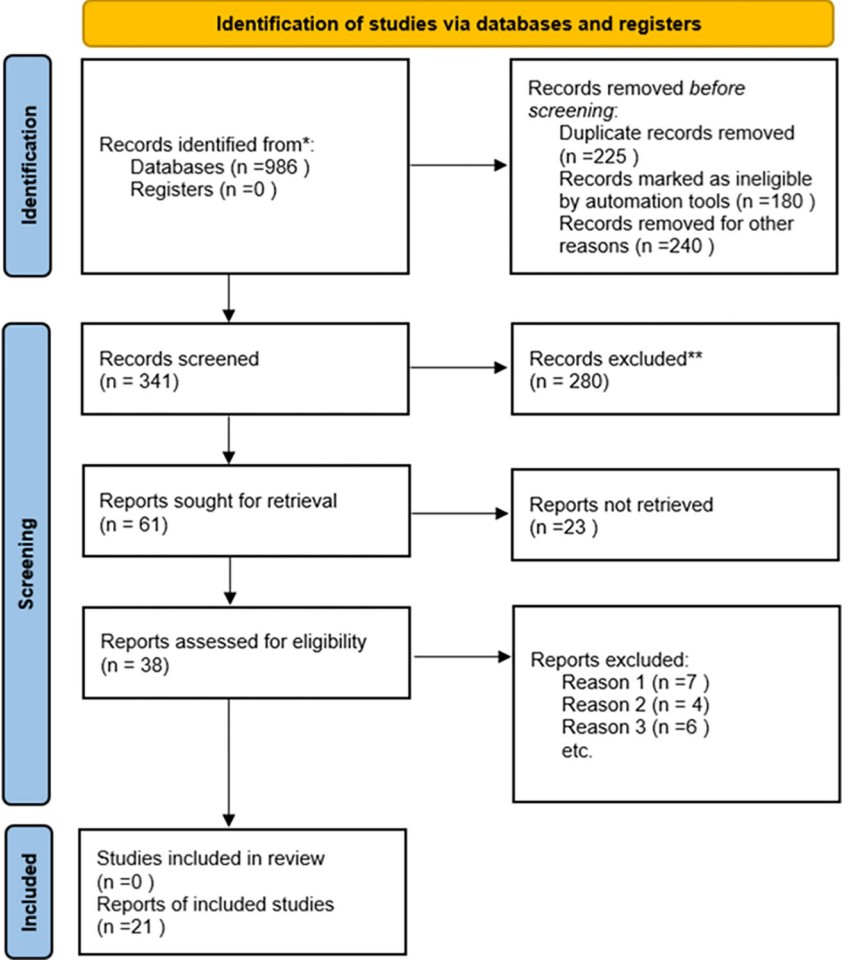

**Fig 1. PRISMA Flowchart for selection of relevant studies.**

hyperlipidemia alone or in combination with diabetes mellitus (DM), ASCVD or combined with DM, acute coronary syndrome (ACS), coronary artery disease, Large-artery atherosclerosis (LAA), cerebral infarction, ST-segment elevation myocardial infarction (STEMI). The interventions were ezetimibe 10 mg plus rosuvastatin 10 mg against rosuvastatin 20 mg, twelve of the included trials were published in Korea, eleven in China, and the treatment ranged in length from 8 weeks to 3 years. According to the Jadad system, 6 studies were categorized as low-quality studies and the other 15 studies were categorized as high-quality studies (Table 1).

## Meta-analysis of long-term composite cardiovascular events

Six RCTs (10 data sets) reported a Primary endpoint, five RCTs (8 data sets) reported a Secondary efficacy endpoint and four RCTs (6 data sets) reported an individual clinical endpoint. We used a randomized controlled model for analysis and discovered that combination therapy was comparable to rosuvastatin monotherapy in preventing Primary endpoint (RR 0.93,95% CI: [0.86, 1.01] p = 0.08), Secondary efficacy endpoint (RR 0.94,95% CI: [0.86, 1.02] p = 0.16) and Individual clinical endpoint (RR 1.01,95% CI: [0.88, 1.16] p = 0.87), with no difference of statistical significance (Fig 2). However, a sensitivity analysis is required due to the large heterogeneity of individual clinical endpoints.

## Meta-analysis of lipid profile

For changes in LDL-C, we included 18 studies (22 data sets) and there was a large heterogeneity among studies ($I^2$ = 61%, P < 0.01). A randomized model was used for the analysis. Meta-analysis showed that the combination group improved LDL-C better than the monotherapy group, and the difference was statistically significant (MD -8.06, 95% CI [-9.48, -6.64] p < 0.05) (Fig 3).

A total of 11 studies (13 data sets) reported changes in HDL-C, TC, and TG. The results showed that moderate-intensity statin plus ezetimibe produced significantly superior reduction than high-intensity statin alone in the TG level (MD -5.66, 95% CI [-8.51, -2.82] p < 0.05), but no significant difference for the effects of treatments on HDL-C (MD 0.24, 95% CI [-1.37, 1.84] p = 0.77) or TC level (MD -13.12 [-27.20, 0.96] p = 0.07) (Fig 3).

Three studies reported non-HDL-C, which we analyzed using a random-effects model, and the results showed that the combination group improved non-HDL-C more than the monotherapy (MD -7.27, 95% CI [-9.80, -4.74] p < 0.05) accompanied by a high degree of heterogeneity ($I^2$ = 97%, P < 0.01), and the difference was statistically significant (Fig 3).

## Meta-analysis of the proportion of patients whose LDL-C levels

Five studies (8 data sets) assessed the proportion of patients who achieved LDL-C levels <70 mg/dL. The results of the test for heterogeneity between studies were P = 0.43 and $I^2$ = 0%, so the fixed model was used for analysis. The results showed that the compliance rate was high in the combined group and the difference was significant (RR 1.26, 95% CI: [1.22, 1.29] p<0.05); similar results were observed in the proportion of patients who achieved 55mg/dL (RR 1.66, 95% CI: [1.56, 1.77] p<0.05) of three studies (5 data sets) included in the total (Fig 4).

## Meta-analysis of overall adverse events

Fourteen studies and 23200 participants were involved in all adverse event assessments and a fixed model was used for the analysis with low heterogeneity among studies ($I^2$ = 43%, P = 0.03), which showed that the rate of adverse events was lower in the combination group than in the high-dose group (RR 0.76, 95% CI: [0.71, 0.81] p<0.05) (Fig 5).

**Table 1. A meta-analysis of the general characteristics of the studies is included.**

| Author year | Country | Trial Duration | Patient Comparison | R10+EZ10 (n) | R20 (n) | Age (years) | | Male (%) | | outcome indicator | Jadad score |
|---|---|---|---|---|---|---|---|---|---|---|---|
| | | | | | | R10+EZ10 | R20 | R10+EZ10 | R20 | | |
| Bomlee 2023[10] | Korea | 3-year | ASCVD | 1894 | 1886 | 63.98 ±10.12 | 64.58 ±10.24 | 1420 (75) | 1406 (74.5) | ①⑦⑧ | 6 |
| Choi2023 [9] | Korea | Week24 | ASCVD | 126 | 132 | 58.93 ±26.99 | 53.66 ±36.73 | 91 (72.2) | 104 (78.8) | ②⑧ | 6 |
| Du2021 [11] | China | 1-year | ASCVD | 35 | 35 | 56.54 ±6.49 | 57.03 ±6.46 | 20 (57.14) | 19 (54.29) | ② | 3 |
| Feng2019 [12] | China | Week24 | CAD | 35 | 34 | 59±9 | 61±8 | 28 (80) | 24 (71) | ②③④⑤⑧ | 3 |
| Hong2018 [13] | Korea | Week 8 | Hypercholesterolemia | 66 | 64 | 62.5 ±8.9 | 64.2±8.3 | 39 (59.1) | 40 (62.5) | ②⑧ | 4 |
| Hyup lee2023(1) [14] | Korea | 3-year | ASCVD | 273 | 301 | 77 ±2 | 77±2 | 173 (63.4) | 180 (59.8) | ①②⑧ | 6 |
| Hyup lee2023(2) [14] | Korea | 3-year | ASCVD | 1621 | 1585 | 61 ± 8 | 62 ±8 | 1247 (76.9) | 1226 (77.4) | ①②⑧ | 6 |
| Joon lee2023(1) [15] | Korea | 3-year | ASCVD+DM | 701 | 697 | 64±9 | 65±9 | 545 (77.7) | 515 (73.9) | ①②③④⑤⑦⑧ | 6 |
| Joonlee2023(2) [15] | Korea | 3-year | ASCVD | 1193 | 1189 | 63±10 | 63±10 | 875 (73.3) | 891 (74.9) | ①②③④⑤⑦⑧ | 6 |
| Kim2016 [16] | Korea | Week 8 | Hypercholesterolemia | 203 | 204 | 64.2±7.9 | 64.3±9.3 | 113 (55.7) | 118 (57.8) | ②③④⑤⑥ | 6 |
| Kim2018[17] | Korea | Week 8 | Hypercholesterolemia | 60 | 63 | 61.77 ±9.92 | 59.33 ±9.13 | 31 (51.7) | 39 (61.9) | ②③④⑤⑥⑧ | 6 |
| Kim2022 [8] | Korea | 3-year | ASCVD | 1894 | 1886 | 64±10 | 64±10 | 1420 (75) | 1406 (75) | ①②⑦⑧ | 5 |
| Kim2023 (1)[18] | Korea | 3-year | ASCVD | 474 | 480 | 67.1±8.4 | 67.8±8.5 | Female | | ①②③④⑤⑦⑧ | 5 |
| Kim2023 (2)[18] | Korea | 3-year | ASCVD | 1420 | 1406 | 62.4±9.6 | 62.8±9.7 | Male | | ①②③④⑤⑦⑧ | 5 |
| Lee2023(1) [19] | Korea | 3-year | ASCVD | 757 | 754 | 63.6 ±9.9 | 64.3±10.3 | 616 (81.4) | 600 (79.6) | ①②⑦⑧ | 6 |
| Lee2023(2)[19] | Korea | 3-year | ASCVD | 1137 | 1132 | 63.5 ±9.3 | 63.9 ±9.2 | 804 (70.7) | 806 (71.2) | ①②⑦⑧ | 6 |
| Li2020[20] | China | Week12 | LAA | 92 | 92 | 73.4±6.25 | 71.0±3.62 | 47 (51.09) | 49 (53.27) | ⑧ | 5 |
| Ma2015 [21] | China | Week16 | Hypercholesterolemia | 40 | 40 | 63. 5±9. 6 | 62. 8±9. 9 | NA | NA | ②③④⑤ | 3 |
| Moon [22] 2023 | Korea | Week24 | ASCVD + DM | 48 | 51 | 61.88 ±6.47 | 61.16 ±7.09 | 28 (58.33) | 37 (72.55) | ②⑧ | 4 |
| Ran2017 [23] | China | Week 12 | ACS | 42 | 41 | 60.4 ±8.2 | 60.5±10.0 | 32 (76.2) | 30 (73.2) | ②③④⑤⑥⑧ | 5 |
| Su2016 [24] | China | Week 12 | CAD | 48 | 48 | 53.18 ±4.32 | 53.2±4.09 | 30 (62.5) | 31 (64.58) | ②③④⑤ | 4 |
| Wang2018[25] | China | Week12 | Hypercholesterolemia + DM | 26 | 26 | 57.3±10.4 | | NA | NA | ②③④⑤⑧ | 3 |

*(Continued)*

**Table 1.** (Continued)

| Author year | Country | Trial Duration | Patient Comparison | R10+EZ10 (n) | R20 (n) | Age (years) | | Male (%) | | outcome indicator | Jadad score |
|---|---|---|---|---|---|---|---|---|---|---|---|
| | | | | | | R10 +EZ10 | R20 | R10 +EZ10 | R20 | | |
| Xu2013 [24] | China | Week16 | cerebral infarction | 29 | 27 | 60±11 | 60±0.9 | 16 (55.2) | 13 (48.1) | ②③④⑤ | 3 |
| Yang2016 [26] | Korea | Week 12 | CAD | 38 | 39 | 62.1 ±9.5 | 62.7 ±9.6 | 24 (63.2) | 26 (66.7) | ②③④⑤ | 5 |
| Zhang2018[27] | China | 1-year | STEMI | 78 | 50 | 61.01±7.57 | 60.9±8.36 | 56 (71. 8) | 34 (68) | ⑧ | 3 |

Abbreviations: R, Rosuvastatin; EZ, ezetimibe; ASCVD, atherosclerotic cardiovascular disease; CAD, coronary artery disease; DM, diabetes mellitus LAA, Large-artery atherosclerosis; ACS, acute coronary syndrome.

STEMI, ST-segment elevation myocardial infarction.

Note: ①long-term composite ACE②LDL-C③HDL-C ④TC ⑤TG ⑥non-HDL-C ⑦the proportion of patients whose LDL-C levels were below 70 or 55 mg/dL ⑧ the occurrence of adverse events.

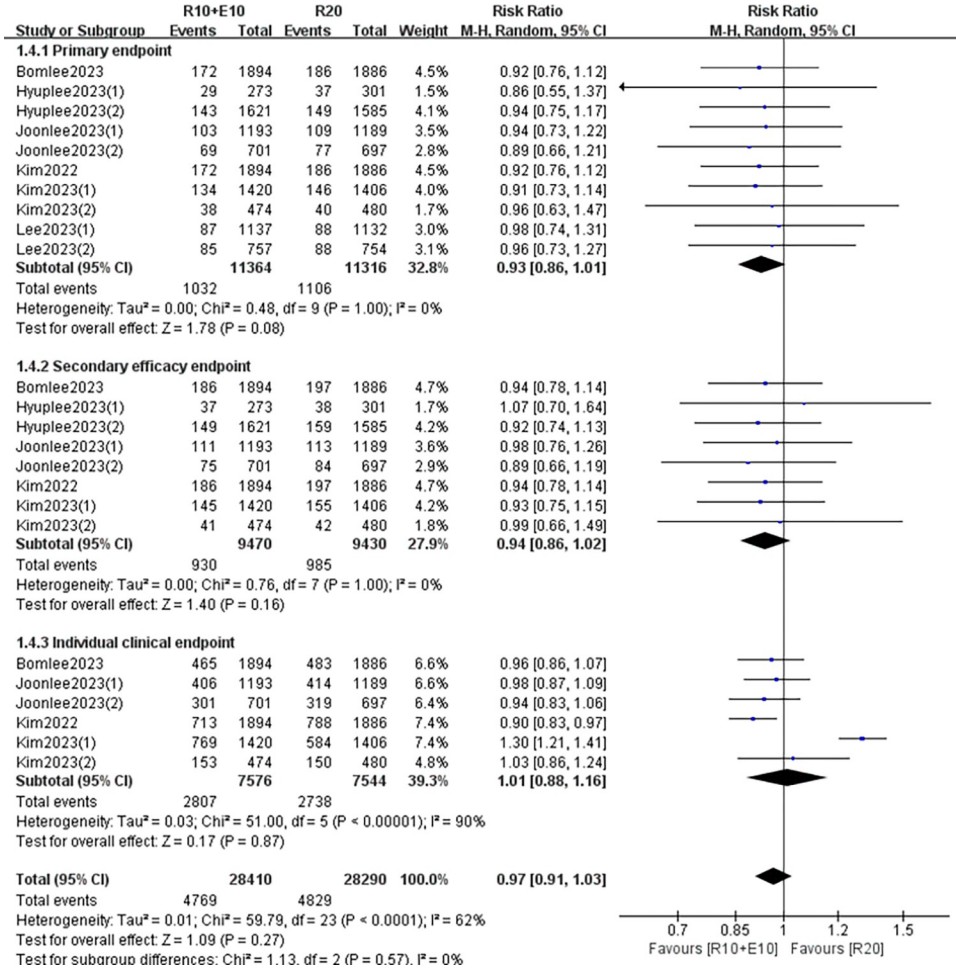

**Fig 2. Forest plot of long-term composite cardiovascular events.**

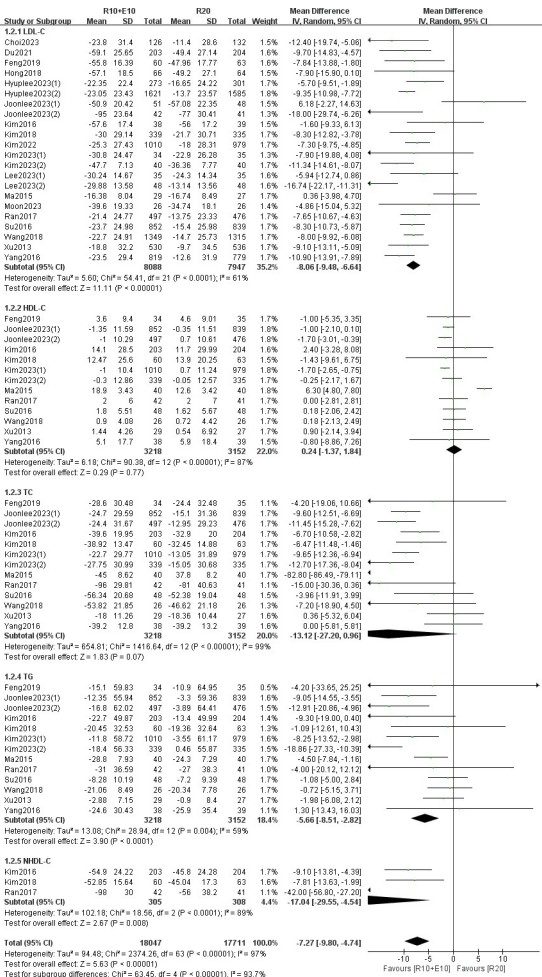

**Fig 3. Forest plot of the results of assessing changes in lipid profile efficacy.**

## Sensitivity analysis

Sensitivity analysis was carried out by excluding individual studies in turn. After excluding literature with high heterogeneity, the LDL-C、TG、NON-HDL-C differences remained statistically significant, indicating that the analysis was relatively stable, but ACE、HDL-C、TC was altered significantly, suggesting that the analysis was unstable (Table 2).

## Publication bias

Five outcome indicators (LDL-C, HDL-C, TC, TG, adverse events) were included in ≥10 studies, and the risk of publication bias was assessed using an inverted funnel plot, which showed a symmetry between the left and right sides of the results, no publication bias and stable results (Figs 6 and 7).

## Discussion

The incidence of CVD is on the rise, imposing a huge burden on society and the economy, and lipid management cannot be delayed. Low control of dyslipidemia and twice as many adverse cardiovascular events compared to those with normal lipids can seriously affect the

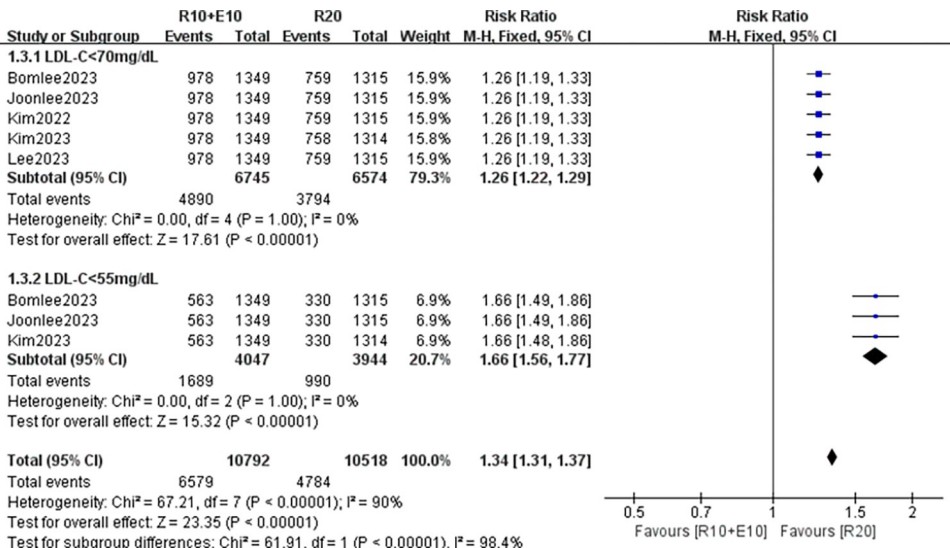

**Fig 4. Forest plot of the proportion of patients who achieved LDL-C levels <70 mg/dL and 55mg/dL.**

quality of patient survival [28,29]. Common lipid markers used to assess CVD risk include TC, LDL-C, HDL-C, TG and non-HDL-C, and in most lipid-lowering intervention studies, LDL-C was the strongest independent predictor of the relationship between the effect of lipid-lowering and the reduction of ASCVD risk. Therefore, most national or regional lipid management guidelines recommend LDL-C as the primary goal of lipid-lowering therapy. The ESC/EAS guidelines for lipid management state that low-density lipoprotein cholesterol (LDL-C) levels should be controlled at less than 55 mg/dL and reduced by at least 50% for patients at very high and high risk of cardiovascular disease, respectively [5,30].

Statins significantly reduce cardiovascular morbidity and mortality while lowering serum cholesterol [31]. Therefore, to regulate lipids up to standard, statins should be preferred clinically, but for patients who do not achieve LDL-C after conventional-dose statin treatment,

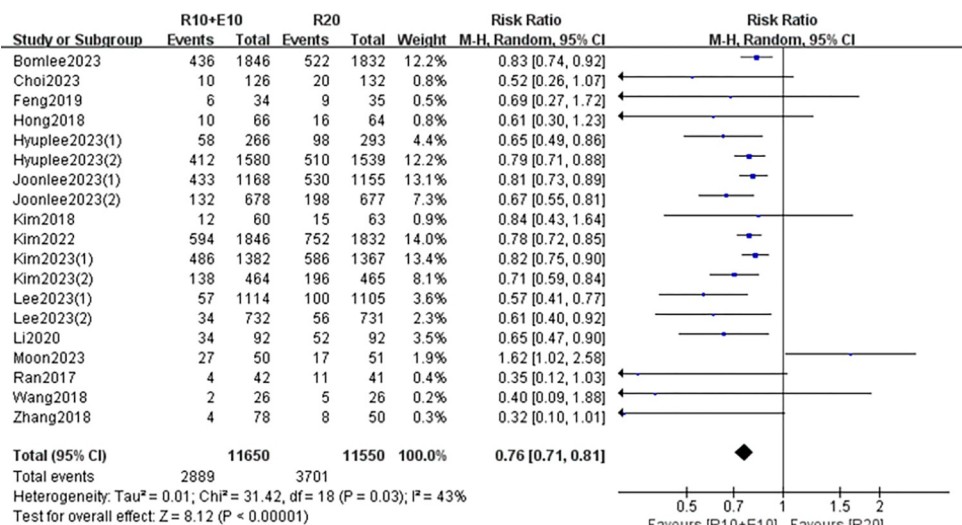

**Fig 5. Forest plot of the results of the occurrence of adverse events.**

**Table 2. Test of heterogeneity publication bias.**

| outcome | No. of study | 95%CI | P value | I²(%) |
|---|---|---|---|---|
| ACE | Kim2023(2)[18] | [0.90, 0.97] | 0.0009 | 0 |
| LDL-C | Ma2015[21] | [-9.80, -7.31] | 0.0001 | 48 |
| HDL-C | Ma2015[21] | [-1.56, -0.49] | 0.0002 | 0 |
| TC | Ma2015[21] | [-9.83, -5.04] | 0.0001 | 61 |
| TG | Kim2023(2)[18] | [-6.84, -2.21] | 0.0001 | 38 |
| non-HDL-C | Ran2017[23] | [-12.25, -4.93] | 0.0001 | 0 |

doubling the statin dose or combining with other lipid-regulating drugs such as ezetimibe may be considered. Some studies have shown that statin combined with ezetimibe reduces LDL-C more significantly than doubling the statin dose, bringing LDL-C to the therapeutic target in the CVD population, but it has also been suggested that combining ezetimibe may not provide as much cardiovascular benefit as doubling the dose [32,33]. In this study, we compared the effects of combining ezetimibe with double-dose rosuvastatin on the risk of long-term composite ACE of the subgroup analyses that combination therapy was comparable to that of monotherapy; after removing the Kim [18] study for sensitivity analyses, the incidence of Individual clinical CVD was found to be lower in the combination therapy group than in the monotherapy group.

Our analysis of lipid parameters and safety showed that combination therapy improved LDL-C, TG, and non-HDL-C and reduced the incidence of adverse reactions more than monotherapy. Also, with combination therapy a superior proportion of patients to monotherapy achieved LDL-C <70 and LDL-C <55. Furthermore, there was no statistical difference between the two in improving HDL-C, and TC. Adding ezetimibe to rosuvastatin therapy is preferable to increasing the statin dose, and it not only lowers LDL-C but also reduces the

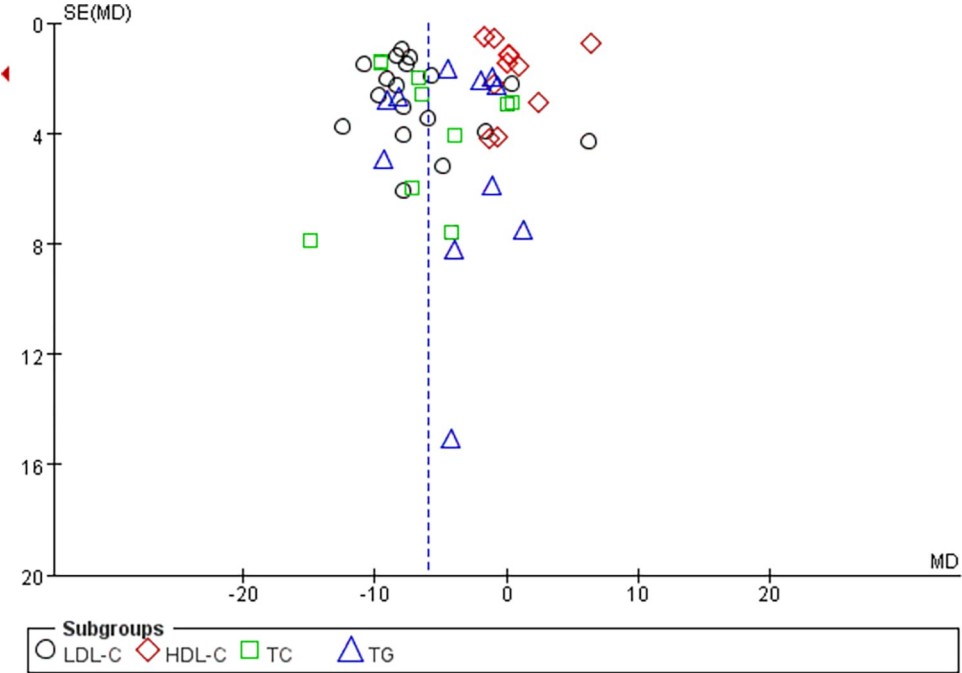

**Fig 6. Funnel diagram of LDL-C, HDL-C, TC, TG.**

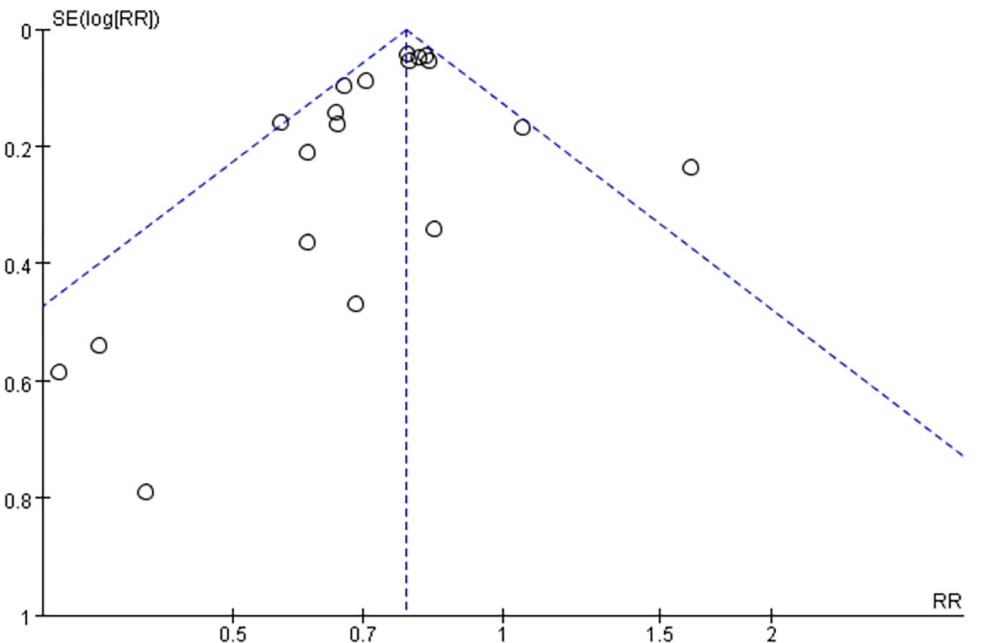

**Fig 7. Funnel diagram of adverse events.**

incidence of adverse reactions. Previous meta-analyses showed that for every 1 mmol/L reduction in LDL-C, ASCVD events were reduced by 20%~23% [34,35]. The outcome successfully demonstrates a strong association between LDL-C and CVD risk; more extensive studies are needed to confirm.

In addition, the beneficial effects of the combination therapy on TG and non-HDL-C were greater compared with high-intensity Rosuvastatin monotherapy, which we consider a noteworthy result. In recent years, the role of non-HDL-C in lipid management for the prevention of ASCVD has become increasingly important, it better anticipates ASCVD risk in patients with obesity, diabetes, and metabolic syndrome [36]. It has been demonstrated that non-HDL-C predicts ASCVD risk better than LDL-C, with or without statin therapy. In a meta-analysis of statin studies, it was found that the magnitude of ASCVD reduction correlated better with the magnitude of non-HDL-C decrease than with the magnitude of LDL-C reduction [37,38]. International guidelines recommend that non-HDL-C should be a co-primary therapeutic target of lipid-lowering therapy for CVD risk reduction, especially for patients at high risk of dyslipidemia [39–42]. NON-HDL-C may be an independent CVD risk factor in the future.

Epidemiological studies have shown that elevated TG levels are a risk factor for CVD, and previous studies have shown that the association between TG and CVD risk is attenuated after adjustment for HDL-C and non-HDL-C, but it is still significant [43,44]. High TG levels are associated with elevated cholesterol and low HLD-C levels, and the American Heart Association (AHA) / American College of Cardiology (ACC) cholesterol guidelines recommend elevated triglycerides as a "risk enhancer" for ASCVD [45]. Our meta-analysis is clinically relevant because we found differences in these lipid levels between the two treatments.

In the heterogeneity test results, the $I^2$ values for LDL-C and TC were 48 and 61, respectively, indicating a moderate degree of heterogeneity (Table 2). Therefore, a reanalysis was performed using the random effects model. The same statistical significance results for LDL-C

indicated the presence of some heterogeneity, but the effect on the meta-analysis was not significant, in contrast to TC, which had a significant effect on the meta-analysis.

Our study has the following limitations: First, some of the study data were converted using formulas, and the data may not be accurate. Second, we included patients with different disease states, which may have contributed to the heterogeneity of this study. Third, we only included studies from China and Korea, and the results are only informative for Asian patients. Despite the limitations, our meta-analysis is meaningful because it provides clinical evidence for better pharmacological treatment of patients with dyslipidemia.

## Conclusions

The study suggests that moderate-intensity Rosuvastatin in combination with ezetimibe can be an alternative to high-intensity statins with better efficacy and safety. Future validation is needed with more long-term high-quality large samples.

## Supporting information

**S1 Table. Supplementary tables.**
(DOC)

**S1 File. Prisma checklist.**
(DOCX)

**S2 File. Literature numbering and literature quality assessment.**
(DOC)

**S3 File. Data for meta-analysis.**
(XLSX)

## Author Contributions

**Conceptualization:** Lingyan Liu, Yongkun Deng, Lei Li, Yong Lai.

**Data curation:** Lingyan Liu, Yongkun Deng, Xingbiao Yang, Yong Lai.

**Formal analysis:** Lingyan Liu.

**Investigation:** Lingyan Liu, Yongkun Deng, Lei Li, Xingbiao Yang, Zhaoheng Yin, Yong Lai.

**Methodology:** Lingyan Liu, Yongkun Deng, Lei Li, Xingbiao Yang, Zhaoheng Yin, Yong Lai.

**Project administration:** Yongkun Deng.

**Resources:** Lingyan Liu, Yongkun Deng, Xingbiao Yang, Zhaoheng Yin, Yong Lai.

**Software:** Xingbiao Yang, Zhaoheng Yin.

**Supervision:** Yongkun Deng, Lei Li, Yong Lai.

**Validation:** Lingyan Liu, Yongkun Deng, Lei Li, Zhaoheng Yin, Yong Lai.

**Visualization:** Lingyan Liu, Yongkun Deng, Lei Li, Xingbiao Yang, Zhaoheng Yin.

**Writing – original draft:** Lingyan Liu.

**Writing – review & editing:** Lingyan Liu, Yongkun Deng, Lei Li, Xingbiao Yang, Zhaoheng Yin, Yong Lai.

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
