## [Decision Letter · Decision Letter 0]

22 Jul 2024

PONE-D-24-07811Efficacy and safety of Moderate-intensity Rosuvastatin Plus Ezetimibe versus High-intensity Rosuvastatin Monotherapy in the treatment of Composite Cardiovascular Events with hypercholesterolemia: A Meta-analysisPLOS ONE

Dear Dr. liu,

Thank you for submitting your manuscript to PLOS ONE. After careful consideration, we feel that it has merit but does not fully meet PLOS ONE’s publication criteria as it currently stands. Therefore, we invite you to submit a revised version of the manuscript that addresses the points raised during the review process.

We look forward to receiving your revised manuscript.

Kind regards,

Jaspinder Kaur, MD

Academic Editor

PLOS ONE

Reviewers' comments:

Reviewer's Responses to Questions

**Comments to the Author**

1. Is the manuscript technically sound, and do the data support the conclusions?

Reviewer #1: Yes

Reviewer #2: Yes

2. Has the statistical analysis been performed appropriately and rigorously? 

Reviewer #1: Yes

Reviewer #2: Yes

3. Have the authors made all data underlying the findings in their manuscript fully available?

Reviewer #1: Yes

Reviewer #2: Yes

4. Is the manuscript presented in an intelligible fashion and written in standard English?

Reviewer #1: Yes

Reviewer #2: Yes

5. Review Comments to the Author

Reviewer #1: I have suggested only few grammatical mistakes that needs to be corrected over all very well written analysis on RCT comparing statin therapy and ezetimibe I do advise to correct the first opening sentence

Reviewer #2: This study addresses a critical gap in current clinical practice by systematically comparing the lipid-lowering efficacy and safety of moderate-intensity rosuvastatin combined with ezetimibe versus high-intensity rosuvastatin monotherapy. Through a rigorous meta-analysis of randomized controlled trials (RCTs), this research aims to inform and refine treatment strategies, enhancing patient outcomes and guiding clinicians in the optimal management of dyslipidemia.

I would propose minor points to be revised:

-Abstract

In the section of results, please replace in the first sentence comma with full stop.

-Introduction

First Paragraph:

Consider revising "considered the cornerstone of control of dyslipidemia or atherosclerotic cardiovascular disease (ASCVD)" to "considered the cornerstone in the control of dyslipidemia and atherosclerotic cardiovascular disease (ASCVD)".

Second Paragraph:

Revise "more studies at both domestic and international" to "more studies, both domestic and international".

Consider adding a comma after "improve the safety during treatment".

Third Paragraph:

Revise "Recently, Various randomized controlled trials" to "Recently, various randomized controlled trials".

-Results

Search Results:

Revise "280 full-text papers were rejected according to the inclusion and exclusion criterion" to "280 full-text papers were excluded based on the inclusion and exclusion criteria".

Study Characteristics and Quality Assessment:

Consider revising "The meta-analysis presented 21 RCTS" to "The meta-analysis included 21 RCTs".

-Discussion

First Paragraph:

Revise "twice as many adverse cardiovascular events as those with normal lipids" to "twice as many adverse cardiovascular events compared to those with normal lipids".

Second Paragraph:

Clarify "most national or regional lipid management guidelines recommend LDL-Cas the primary goal" by adding a space in "LDL-Cas" to "LDL-C as".

6. PLOS authors have the option to publish the peer review history of their article (what does this mean?). If published, this will include your full peer review and any attached files.

Reviewer #1: **Yes: **Gurpreet Kaur Saini

Reviewer #2: No

---

## [Author Response · Author response to Decision Letter 0]

30 Jul 2024

We are grateful to the editor and two reviewers for their time and energy in providing helpful comments that have improved the manuscript. In this revision, we have addressed all of these comments.

In this document, we describe how we have addressed the reviewers’ comments. Referee comments are shown in black italics and author responses are shown in blue regular text. A manuscript with tracking changes is attached separately.

Editorial Request: 

Comment 1:Please ensure that your manuscript meets PLOS ONE's style requirements, including those for file naming.

Response 1: Thank you for your detailed comments. We have ensured that our manuscript meets the style requirements and file naming conventions of PLOS ONE.

Comment 2: We note that your Data Availability Statement is currently as follows: [All relevant data are within the manuscript and its Supporting Information files.]

Please confirm at this time whether or not your submission contains all raw data required to replicate the results of your study. Authors must share the “minimal data set” for their submission. PLOS defines the minimal data set to consist of the data required to replicate all study findings reported in the article, as well as related metadata and methods.

Response 2:Thank you for your valuable advice.We have confirmed that our submission contains the values behind the means, standard deviations and other measures reported.

Comment 3: Please review your reference list to ensure that it is complete and correct. If you have cited papers that have been retracted, please include the rationale for doing so in the manuscript text, or remove these references and replace them with relevant current references. Any changes to the reference list should be mentioned in the rebuttal letter that accompanies your revised manuscript. If you need to cite a retracted article, indicate the article’s retracted status in the References list and also include a citation and full reference for the retraction notice.

Response 3: Thank you very much for your valuable suggestions. According to your suggestions we have checked and unified references for consistent formatting.

Reviewer #1 

Comment 1: I have suggested only few grammatical mistakes that needs to be corrected over all very well written analysis on RCT comparing statin therapy and ezetimibe I do advise to correct the first opening sentence."The gold standard in the treatment of dyslipidemia, statins significantly reduce the risk of cardiovascular disease". statins are the gold standard in the treatment better start sentence this way .

Response 1: We would like to thank the reviewer for this suggestion. We have revised the language's grammatical and syntax errors.Revising "The gold standard in the treatment of dyslipidemia, statins significantly reduce the risk of cardiovascular disease" to "Statins are the gold standard in the treatment of dyslipidemia, significantly reducing the risk of cardiovascular disease."(Line 24-25)

Comment 2:-Abstract: In the section of results, The meta-analysis described 21 RCTs involving 24592 participants, The findings indicated that combination therapy,in this RCTs does the combination therapy was high intensity statin or moderate ?

Response 2: Thank you for your suggestion.Revising "The meta-analysis described 21 RCTs involving 24592 participants. The findings indicated that combination therapy" to "The meta-analysis described 21 RCTs involving 24592 participants. The findings indicated that moderate-intensity statin combination therapy".(Line 41)

Comment 3:-Introduction:First Paragraph: several statins are available of which rosuvastatin “pls add word of”

Response 3:Thank you for your suggestion.Revising "Several statins are available, which " to "Several statins are available of which ".(Line 67)

Comment 4:-Introduction:Third Paragraph: “Various” use lower case for Various 

Response 4:We would like to thank the reviewer for this suggestion.Revising "Various " to "various".(Line 81)

Comment 5:-Discussion First Paragraph: “LDL-Cas” space between LDL C and as the primary 

Response 5:We would like to thank the reviewer for this suggestion.Revising "LDL-Cas " to "LDL-C as".(Line 254)

Reviewer #2: This study addresses a critical gap in current clinical practice by systematically comparing the lipid-lowering efficacy and safety of moderate-intensity rosuvastatin combined with ezetimibe versus high-intensity rosuvastatin monotherapy. Through a rigorous meta-analysis of randomized controlled trials (RCTs), this research aims to inform and refine treatment strategies, enhancing patient outcomes and guiding clinicians in the optimal management of dyslipidemia.

I would propose minor points to be revised:

Comment 1: -Abstract: In the section of results, please replace in the first sentence comma with full stop.

Response1: Thank you for your valuable comments.We have revised and replace in the first sentence comma with full stop. (Line 25)

Comment 2: -Introduction: First Paragraph: Consider revising "considered the cornerstone of control of dyslipidemia or atherosclerotic cardiovascular disease (ASCVD)" to "considered the cornerstone in the control of dyslipidemia and atherosclerotic cardiovascular disease (ASCVD)".

Response 2: Thank you for your suggestions, we have revised it.(Line 65-66)

Comment 3: Second Paragraph: Revise "more studies at both domestic and international" to "more studies, both domestic and international”. Consider adding a comma after "improve the safety during treatment".

Response 3: Thank you for your suggestions, we have revised it.(Line 73-74)

Comment 4: Third Paragraph: Revise "Recently, Various randomized controlled trials" to "Recently, various randomized controlled trials".

Response 4: Thank you for your suggestions, we have revised it.(Line 81)

Comment 5: -Results Search Results: Revise "280 full-text papers were rejected according to the inclusion and exclusion criterion" to "280 full-text papers were excluded based on the inclusion and exclusion criteria".

Response 5: Thank you for your suggestions, we have revised it.(Line150-151)

Comment 6: Study Characteristics and Quality Assessment: Consider revising "The meta-analysis presented 21 RCTS" to "The meta-analysis included 21 RCTs".

Response 6: Thank you for your suggestions, we have revised it.(Line 157)

Comment 7: -Discussion First Paragraph: Revise "twice as many adverse cardiovascular events as those with normal lipids" to "twice as many adverse cardiovascular events compared to those with normal lipids".

Response 7: Thank you for your suggestions, we have revised it.(Line 247)

Comment 8 Second Paragraph: Clarify "most national or regional lipid management guidelines recommend LDL-Cas the primary goal" by adding a space in "LDL-Cas" to "LDL-C as".

Response 8: Thank you for your suggestions., we have revised it.(Line 253)

---

## [Decision Letter · Decision Letter 1]

23 Aug 2024

Efficacy and safety of moderate-intensity rosuvastatin plus ezetimibe versus high-intensity rosuvastatin monotherapy in the treatment of composite cardiovascular events with hypercholesterolemia: A meta-analysis

PONE-D-24-07811R1

Dear Dr. liu,

We’re pleased to inform you that your manuscript has been judged scientifically suitable for publication and will be formally accepted for publication once it meets all outstanding technical requirements.

Kind regards,

Jaspinder Kaur, MD

Academic Editor

PLOS ONE

Additional Editor Comments (optional):

Reviewers' comments:

Reviewer's Responses to Questions

**Comments to the Author**

1. If the authors have adequately addressed your comments raised in a previous round of review and you feel that this manuscript is now acceptable for publication, you may indicate that here to bypass the “Comments to the Author” section, enter your conflict of interest statement in the “Confidential to Editor” section, and submit your "Accept" recommendation.

Reviewer #2: All comments have been addressed

2. Is the manuscript technically sound, and do the data support the conclusions?

Reviewer #2: Yes

3. Has the statistical analysis been performed appropriately and rigorously? 

Reviewer #2: Yes

4. Have the authors made all data underlying the findings in their manuscript fully available?

Reviewer #2: Yes

5. Is the manuscript presented in an intelligible fashion and written in standard English?

Reviewer #2: Yes

6. Review Comments to the Author

Reviewer #2: All comments were answered one by one in the revised form of manuscript by indicating the exact line.

7. PLOS authors have the option to publish the peer review history of their article (what does this mean?). If published, this will include your full peer review and any attached files.

Reviewer #2: No

---

## [Editor Report · Acceptance letter]

8 Sep 2024

PONE-D-24-07811R1 

PLOS ONE

Dear Dr. Liu, 

I'm pleased to inform you that your manuscript has been deemed suitable for publication in PLOS ONE. Congratulations! Your manuscript is now being handed over to our production team.

Kind regards, 

on behalf of

Dr. Jaspinder Kaur 

Academic Editor

PLOS ONE